# Immunogenomic profiling determines responses to combined PARP and PD-1 inhibition in ovarian cancer

Anniina Färkkilä [1,2,3,4], Doga C. Gulhan[2,11], Julia Casado [3,11], Connor A. Jacobson [4], Huy Nguyen [1], Bose Kochupurakkal[1], Zoltan Maliga[4], Clarence Yapp [4], Yu-An Chen [4], Denis Schapiro[4], Yinghui Zhou[5], Julie R. Graham [5], Bruce J. Dezube[5], Pamela Munster [6], Sandro Santagata [7], Elizabeth Garcia[8], Scott Rodig[8], Ana Lako[8], Dipanjan Chowdhury [1], Geoffrey I. Shapiro[1], Ursula A. Matulonis[1], Peter J. Park [9], Sampsa Hautaniemi [3], Peter K. Sorger [4], Elizabeth M. Swisher [10], Alan D. D'Andrea [1,11✉] & Panagiotis A. Konstantinopoulos[1,11✉]

Combined PARP and immune checkpoint inhibition has yielded encouraging results in ovarian cancer, but predictive biomarkers are lacking. We performed immunogenomic profiling and highly multiplexed single-cell imaging on tumor samples from patients enrolled in a Phase I/II trial of niraparib and pembrolizumab in ovarian cancer (NCT02657889). We identify two determinants of response; mutational signature 3 reflecting defective homologous recombination DNA repair, and positive immune score as a surrogate of interferon-primed exhausted CD8 + T-cells in the tumor microenvironment. Presence of one or both features associates with an improved outcome while concurrent absence yields no responses. Single-cell spatial analysis reveals prominent interactions of exhausted CD8 + T-cells and PD-L1 + macrophages and PD-L1 + tumor cells as mechanistic determinants of response. Furthermore, spatial analysis of two extreme responders shows differential clustering of exhausted CD8 + T-cells with PD-L1 + macrophages in the first, and exhausted CD8 + T-cells with cancer cells harboring genomic *PD-L1* and *PD-L2* amplification in the second.

[1] Dana-Farber Cancer Institute, 450 Brookline Avenue, Boston, MA 02215, USA. [2] Harvard Medical School, 25 Shattuck Street, Boston, MA 02115, USA. [3] Research Program in Systems Oncology, University of Helsinki, Haartmaninkatu 8, 00014 Helsinki, Finland. [4] Laboratory of Systems Pharmacology, Harvard Medical School, Boston200 Longwood AvenueMA 02115, USA. [5] TESARO: A GSK company, 1000 Winter Street, Waltham, MA 02451, USA. [6] Helen Diller Family Comprehensive Cancer Center, 1450 3rd Street, San Francisco, CA 94158, USA. [7] Brigham and Women's Hospital, Laboratory for Systems Pharmacology, 75 Francis Street, Boston, MA 02115, USA. [8] Department of Pathology, Brigham and Women's Hospital, Harvard Medical School, 75 Francis Street, Boston, MA 02115, USA. [9] Department of Biomedical Informatics, Harvard Medical School, 25 Shattuck Street, Boston, MA 02115, USA. [10] University of Washington, 1959 NE Pacific Street, Seattle, WA 98195, USA. [11]These authors contributed equally: Doga C. Gulhan, Julia Casado, Alan D. D'Andrea, Panagiotis A. Konstantinopoulos. ✉email: Alan_dAndrea@dfci.harvard.edu; Panagiotis_konstantinopoulos@dfci.harvard.edu

Ovarian cancer remains the most lethal gynecologic malignancy and the fifth most frequent cause of cancer-related mortality in women in the US[1]. While chemotherapy, Poly-ADP Ribose Polymerase inhibitor (PARPi) therapy, and antiangiogenic therapy have demonstrated excellent activity in this disease, ovarian cancer is one of the few malignancies where immunotherapy with immune checkpoint blockade exhibits only modest activity [objective response rate (ORR) of ~8–9%] with infrequent durable responses, and currently has no FDA approved indication[2,3]. The development of new strategies for improving the efficacy of immune checkpoint blockade is therefore a high priority for treatment of ovarian cancer.

Preclinical work in murine ovarian cancer models by us and others has demonstrated synergistic antitumor activity for combinations of PARP inhibitors and anti-PD-1/PD-L1 agents[4–6]. Specifically, in a syngeneic genetically engineered mouse model of high-grade serous ovarian cancer driven via concurrent loss of p53 and Brca1 and overexpression of c-Myc, PARP inhibitor olaparib induced activation of Stimulator of Interferon Genes (STING) pathway accompanied by increased expression of Interferon-beta, PD-L1 and CXCL10[4]. In the same model, combination therapy with olaparib and PD-1 blockade augmented the activity of olaparib while anti-PD-1 alone did not have any effect. Similar results, but using a homologous recombination proficient ID8 model, were reported by Shen et al.[5], which showed that the PARP inhibitor talazoparib induced STING activation, increased PD-L1, CCL5, and CXCL10 expression and exhibited synergistic activity with an anti-PD-L1 antibody.

Based on this, we conducted a phase I/II clinical trial of the PARPi niraparib in combination with the anti-PD-1 antibody pembrolizumab in recurrent ovarian cancer (TOPACIO trial[7]); the trial enrolled 62 patients with a median of 3 prior lines of therapy (range 1–5). The majority (76%) of patients had acquired platinum resistant or refractory disease while the remaining 24% of patients were ineligible to receive platinum therapy due to prior toxicity or allergic reaction. The niraparib/pembrolizumab combination was well tolerated and exhibited a confirmed ORR of 18% [5% complete responses (CRs) and 13% partial responses (PRs)] and a clinical benefit rate of 65%, clearly exceeding the expected activity of niraparib or pembrolizumab as monotherapies in recurrent platinum-resistant ovarian cancer. Responses were durable and median duration of response was not reached (range 4.2–14.5+) months.

Despite this promising activity, many patients did not respond, highlighting the need for predictive biomarkers to identify ovarian cancer patients prospectively who would benefit from the niraparib/pembrolizumab combination. This is particularly relevant for patients with platinum-resistant ovarian cancer who have poor prognosis and therefore require careful selection of their next treatment regimen. In the context of the TOPACIO trial, known biomarkers of response to PARPi and immune checkpoint blockade were not associated with response to the niraparib/pembrolizumab combination; these biomarkers include tumor BRCA mutation status, homologous recombination deficiency (HRD) status (assessed by the Myriad HRD test), and PD-L1 status. Here, using advanced genomic analyses and single-cell imaging, we show that mutational signature 3 and interferon signaling in the CD8+ T-cell compartment of the tumor microenvironment determine responses to niraparib plus pembrolizumab in patients enrolled in the TOPACIO trial.

## Results

**Mutational signature 3 correlates with clinical benefit.** As earlier analyses had failed to identify an association between the BRCA mutation status and HRD status as assessed by the Myriad

HRD and clinical response[7], we explored alternative determinants of HRD. The clinical characteristics and correlative analyses are summarized in Table 1. First, we performed BROCA targeted sequencing using a panel of 84 DNA repair genes complimented by methylation analysis for BRCA1 and RAD51C (Fig. 1a). BROCA sequencing identified 21/52 (40%) of the patients as HRD. Fourteen of the patients had tumors that were positive for BROCA but negative for BRCA mutations. Eleven of these tumors had BRCA1 hypermethylation, two had mutations in CDK12, and one had RAD51C hypermethylation. Similar to our prior results with other biomarkers of HRD, BROCA status did not associate with response (Fig. 1b and Supplementary Fig. 1A). We also evaluated RAD51 by immunohistochemistry (IHC) as a functional marker for HR deficiency[8]. In total, 11/38 (29%) of the tumors lacked RAD51 foci, and therefore predicted to be HRD (Fig. 1a). However, RAD51 status did not, significantly correlate with response (Fig. 1c and Supplementary Fig. 1 B).

To look for additional genomic markers for HRD and response, we performed OncoPanel targeted sequencing for 447 cancer-related genes (Supplementary Table 1). All tumors were mutated for TP53, and the results of the mutational and copy number variation analyses are summarized in Supplementary Table 3. Of note, tumor mutational burden previously reported to correlate with response to immune checkpoint blockade, was not associated with response to niraparib/pembrolizumab or the other determinants of HRD (data provided in Supplementary Tables 3 and 4). Furthermore, none of the tumors were

**Table 1 Summary of clinical data (A) and patient numbers in correlative analyses (B).**

**(A)**

| Clinical details | N (%) |
|---|---|
| Age (years)[a] | 60 (46–83)[b] |
| ECOG[a] | |
| 1 | 44 (71) |
| 2 | 18 (29) |
| Platinum response[c] | |
| Sensitive/ineligible | 16 (25) |
| Resistant/refractory | 46 (74) |
| Prior lines of therapy[b] | 3 (1–5) |
| Confirmed BOR | |
| CR | 3 (5) |
| PR | 8 (13) |
| SD | 28 (45) |
| PD | 20 (32) |
| ND | 3 (5) |
| Duration of response (days) | 190 (123–441)[b] |

**(B)**

| Correlative analyses | N (%) |
|---|---|
| HRD test | 55 (89) |
| BRCAmut | 60 (97) |
| PD-L1 IHC | 44 (71) |
| RAD51 IHC | 38 (61) |
| BROCA | 52 (84) |
| Oncopanel | 39 (63) |
| Nanostring | 44 (71) |
| CycIF | 26 (37) |

ECOG Eastern Cooperative Oncology Group Performance status, BOR best objective response, CR complete response, PR partial response, SD stable disease, PD progressive disease, ND not defined.
[a]At screening.
[b]Median (range).
[c]Response to last platinum-based chemotherapy.
ECOG; Eastern Cooperative Oncology Group Performance status, BOR; Best objective response, CR; Complete response, PR; Partial response, SD; Stable disease, PD; Progressive disease, ND; Not defined.

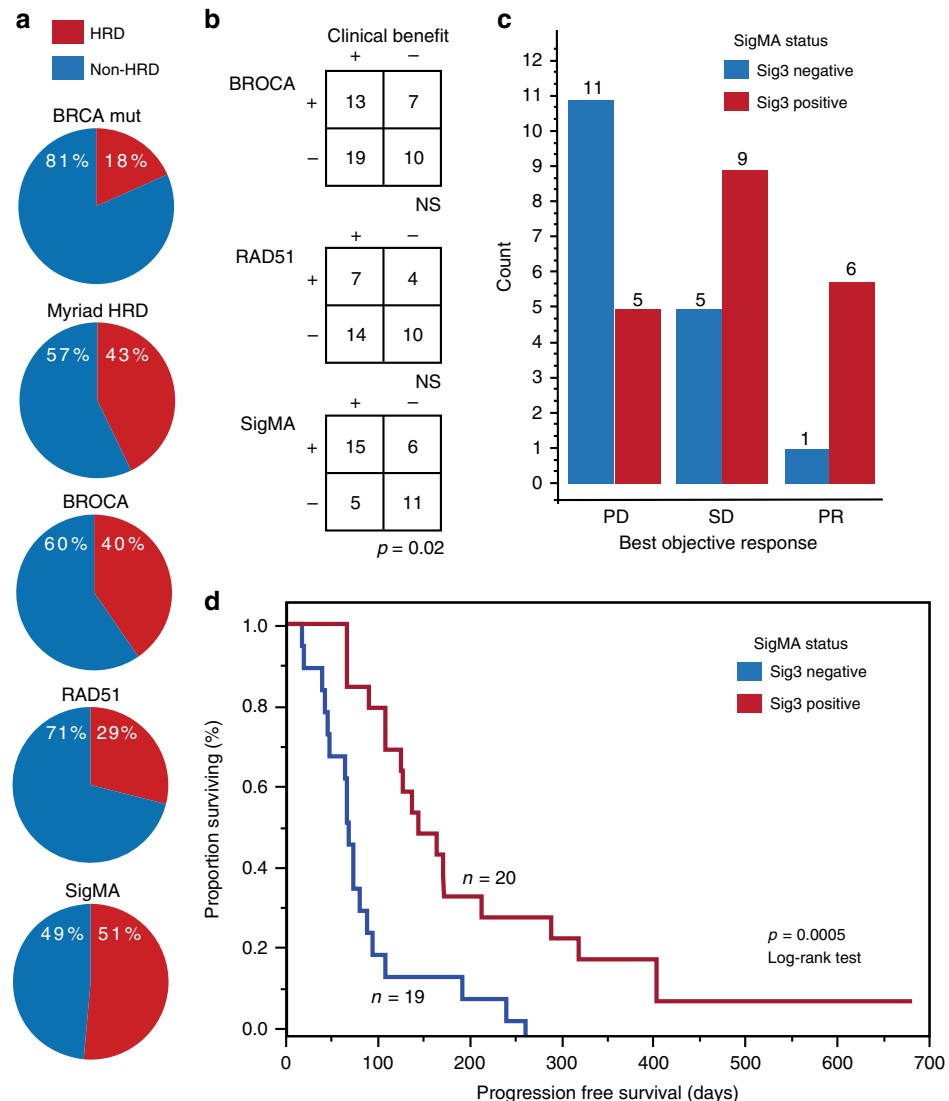

**Fig. 1 Tumor mutational signature 3 positivity associates with prolonged progression-free survival with the combination of niraparib and pembrolizumab. a** SigMA identified a larger proportion of tumors positive for homologous recombination deficiency (HRD). The proportions of tumors positive (red) and negative (blue) for HRD as annotated by the *BRCA*1/2 mutation, Myriad HRD test, BROCA, *RAD51*, and SigMA. **b** Sig3 positivity is associated with clinical benefit as determined by either complete or partial response or stable disease. Correlations of HRD to clinical benefit (Fisher's exact test). **c** Proportions of patients positive (red) or negative (blue) for Sig3 according to best objective response. PD progressive disease, SD Stable disease, PR partial response. **d** Sig3 associates with increased progression-free survival (PFS; Log-rank test). Kaplan–Meier graph for PFS for the combination of niraparib and pembrolizumab according to Sig3 status. All test were two-sided.

polymerase epsilon (*POLE*)-mutated or mismatch repair deficient (MMRD); both are DNA repair defects known to associate with excellent response to immune checkpoint blockade[9,10].

Using a recently developed computational tool called SigMA (Signature Multivariate Analysis), which is capable of determining mutational signatures even when tumor mutational burden is low[11] we were able to test for the presence of a specific mutational signature, which has been shown to be associated with HRD, called signature 3[12]. Using SigMA on the OncoPanel sequencing data, we identified the presence of signature 3 in 51% (20/39) of the patients, thereafter denoted as Sig3 positive. SigMA thus identified a larger proportion of tumors as HRD compared to other markers of HRD (Fig. 1). Of the six tumors with a BRCA mutation, SigMA was positive in four (i.e., consistent with the false-positive rate of 2%; see methods); of note, the two BRCA mutated tumors that were SigMA negative were also BROCA

negative. Furthermore, among the 15 patients with loss-of-function of *BRCA1/2* (nine tumors with BRCA1 hypermethylation, five tumors with BRCA1 mutation and one tumor with BRCA2 mutation), SigMA identified ten to be Sig3 positive, again consistent with the reported sensitivity of the SigMA algorithm (see methods).

We found that Sig3 positivity was indicative of clinical benefit; significantly more Sig3-positive patients had stable disease or partial response ($p = 0.02$, Fisher's exact test), compared to patients who had progressive disease (Fig. 1b, c). Additionally, Sig3 was associated with prolonged progression-free survival (PFS; Fig. 1d): the median PFS in Sig3-positive patients was 5.0 months (range 2.1–22.7) compared to 2.2 months (range 0.5–8.6) in the Sig3-negative patients ($p = 0.0005$, Log-rank test), with a hazard ratio for progression of 0.37 in the Sig3-positive patients (95%CI 0.17–0.80).

**Immune score and Sig3 identify all objective responders**. To profile the immune microenvironment, we performed Nano-String gene expression profiling using the PanCancer IO 360 Gene Expression Panel plus 30 DNA repair genes (Supplementary Table 1). We observed notable differences in the immune microenvironment gene expression patterns between samples obtained from tumors at diagnosis (chemo-naive samples) versus the samples obtained from tumors previously exposed to platinum-based chemotherapy (chemo-exposed samples; Supplementary Fig. 2A, B; see methods). Specifically, chemo-exposed samples exhibited higher scores for immune-related pathways (Supplementary Fig. 2B) and immune cell-type scores (Supplementary Fig. 2C) and had a positive correlation to PD-L1 positivity by immunohistochemistry (Supplementary Fig. 2D). Given these differences between the chemo-naive and chemo-exposed samples, we evaluated the associations to clinical response separately within the chemo-naive or the chemo-exposed samples.

In chemo-naive samples, pathway analysis revealed six pathways significantly enriched among tumors with objective response to niraparib/pembrolizumab (Fig. 2a). Of these pathways, three were related to Type-I interferon signaling and these were also significantly associated with objective response rate (ORR; Fig. 2b). Of note, all responders had a high score (the highest 25% as being positive) for at least one of the 3 Type-I interferon pathways (Supplementary Fig. 2E).

In the chemo-exposed samples, we observed elevated relative scores for exhausted CD8 + T-cells in tumors with an objective response (Fig. 2c). Furthermore, the relative cell-type score for exhausted CD8 + T-cell (calculated using gene expression levels of *CD244, EOMES, LAG3*, and *PTGER4;* see methods) vs. total CD8 + T-cells (calculated using *CD8a, CD8b*) was significantly higher in the responders as compared to non-responders ($p =$ 0.02, Mann–Whitney $U$-test, Fig. 2d). In the subset of patients displaying clinical benefit the relative cell-type score positively correlated with the amount of tumor regression. However, the cell-type scores for the total CD8 + T-cell score or the relative cell-type score, calculated based on the gene expressions for exhausted CD8 + T-cells vs. the gene signatures score for total tumor infiltrating lymphocytes (TILs) were not significantly different in the responders compared to non-responders (Fig. 2d). There were no significant differences between the responders and non-responders in the cell-type scores within the chemo-naive-or pathway scores within the chemo-exposed samples.

We next assigned an Immune Score (IS), annotating the samples with the highest 25% of the pathway score for any of the interferon pathways as being positive for the IS in the chemo-naive, and the samples with the highest 25% of the exhausted CD8 + T-cell/CD8 + T-cell score in the chemo-exposed samples, respectively. IS positivity significantly correlated with OR ($p =$ 0.01, Fisher's exact test, Fig. 2e). As displayed in the waterfall plot, we confirmed that all patients that had an OR to the combination, were positive for either Sig3 or IS or both (Fig. 2f). Positivity for Sig3, IS, or both, defined as being positive for a Combined score, significantly associated with clinical benefit (Fig. 2g), and positivity for the combined score significantly associated with prolonged PFS (Fig. 2h) with a hazards ratio (HR) of 0.32 (95%CI 0.15–0.70, $p =$ 0.002, Log-rank test). Importantly, none of the patients whose tumors were negative for the combined score exhibited an OR to niraparib/pembrolizumab (ORR 0%; Fig. 2i).

**Single-cell imaging reveals potential mechanisms of response**. To gain further insights into the mechanisms of response to combined niraparib/pembrolizumab, we performed highly multiplexed single-cell imaging of 26 tumor samples using t-CyCIF[13]. We generated single-cell resolution data on samples for 30

different antigens (see Supplementary Table 2 for a list of antibodies and their targets), which made it possible to identify immune cell types and assay functional cell states. In total, we analyzed 6.6 million successfully segmented single cells (average $2.5 \times 10^5$ per tumor sample, range $1.6 \times 10^3$–$7.1 \times 10^5$). The mean levels of markers across all cells in the samples did not associate with treatment response or other clinical features (Fig. 3a). We next annotated the cells into pre-defined cell types based on marker expression using FlowSOM (see Methods), and visualized the data using semi-supervised Uniform Manifold Approximation and Projection (UMAP) dimensionality reduction (Fig. 3b). The UMAP embedding and visualization showed agreement with the classified tumor, immune, and stromal cells (top), and further to immune cell subpopulations (bottom), allowing us to interrogate the tumor microenvironment at single-cell resolution.

Consistent with mRNA data, prior exposure to chemotherapy resulted in a lower proportion of tumor cells, and a larger proportion of immune, and stromal cells as compared to the chemo-naive samples (Fig. 3c and Supplementary Fig. 3A–C). Macrophages were the most abundant immune cell-type in the tumor microenvironment followed by CD8 + and CD4 + T-cells (Supplementary Fig. 3F) and both antigen presenting cells and neutrophils were more abundant in chemo-exposed samples (Supplementary Fig. 3D, E). However, differences in tumor microenvironment composition did not associate with response (Supplementary Fig. 3F). At single-cell resolution, PD-L1 expression was the highest in cells corresponding to the macrophage—dendritic cell cluster (Supplementary Fig. 4A), and the expression of PD-L1 was higher only in the stromal cells of chemo-exposed as compared to chemo-naive samples (Supplementary Fig. 4B).

In light of our expression profiling findings showing that Type-I interferon signaling was enriched in samples from patients with responses to niraparib/pembrolizumab, we assessed the cell-type context for interferon pathway activation in the tumor microenvironment using phospho-STAT1 (pSTAT1) expression as a marker for interferon activation. In this regard, increased mean pSTAT1 protein expression levels in exhausted, PD-1 high expressing, CD8 + T-cells significantly associated with OR ($p =$ 0.04, Mann–Whitney $U$-test, Fig. 3d) and clinical benefit ($p =$ 0.03, Mann–Whitney $U$-test, Fig. 3e). Further, increased pSTAT1 and a higher expression of Ki67, indicative of a higher proliferative state, in CD8 + effector T-cells, associated with objective response ($p =$ 0.04 and 0.002, respectively; Mann–Whitney $U$-test; Fig. 3e, f). Higher pSTAT1 and Ki67 was also verified in visualizing the markers in representative immunofluorescence images of a responder compared to a non-responder (Fig. 3h). Interestingly, tumors that were Sig3 positive had higher mean levels of PD-L1 specifically in tumor cells (Supplementary Fig. 4C), and in IBA1 + CD11b + macrophages (Supplementary Fig. 4D), compared to Sig3-negative tumors. These differences in the tumor microenvironment associated with HRD as measured by Sig3, could potentially contribute to the clinical benefit associated with a Sig3-positive status. This association was not detected with the other measures for HRD, including tumor BRCA, Myriad HRD, BROCA, and RAD51 (data provided in Supplementary Table 4). Further, in Cox regression analysis, including relevant clinical and correlative variables, Sig3 was an independent predictor of PFS (Supplementary Table 5).

**Unique single-cell spatial features of extreme responders**. We next performed a deeper phenotyping of samples from two patients with exceptional responses. The clinical details and tumor characteristics of these patients are summarized in Supplementary Table 6: both had platinum-resistant ovarian cancer

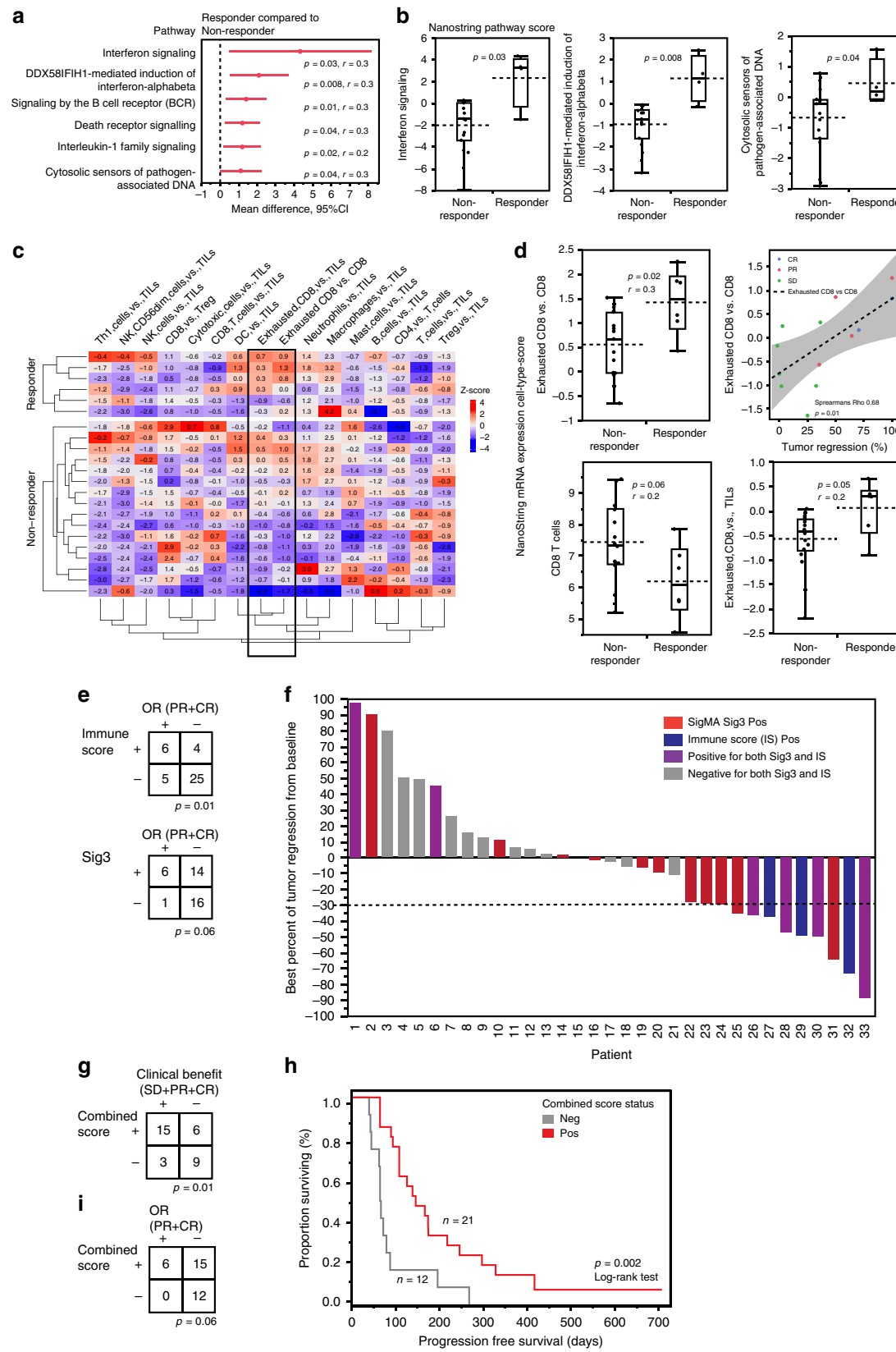

and experienced long-term partial responses, which were still ongoing at the time of the data cutoff (September 2018). The first patient had a PR lasting over 10 months, and an 87% tumor regression from the baseline (Fig. 4a). Her tumor was positive for mutational Sig3, and BROCA analysis revealed a

hypermethylation of *BRCA1* and a loss-of-function mutation in *TP53*. Her tumor was enriched for CD163 + IBA1 + CD11b + macrophages and exhausted CD8 + T-cells (Fig. 4b). The highest PD-L1 expression was observed in CD163 + IBA1 + CD11b + macrophages (Fig. 4c, d). When we looked at the spatial

**Fig. 2 Immune signatures associate with tumor regression and objective response to niraparib and pembrolizumab. a** In the chemo-naive samples, pathway scores for six biological pathways were higher in the responders. The red dots depict the mean difference of Nanostring pathway scores between responders ($n = 4$) compared to non-responders ($n = 15$), and the red lines show the 95% confidence interval (CI) for the difference. $p < 0.05$ was considered significant, $r$; effect size, Mann–Whitney $U$-test. **b** The Nanostring pathway scores of three pathways related to Type-I interferon signaling were significantly higher in the responders compared to non-responders (Mann–Whitney $U$-test). **c** In the chemo-exposed samples, heatmap of the Nanostring cell-type $z$-scores showed a higher signature of relative exhausted CD8 + T-cell-scores (black box) in the responders (CR, PR; $n = 6$) compared to non-responders (SD, PD; $n = 16$). **d** The inferred relative cell-type scores from Nanostring advanced analysis are presented as the specific cell-type score over (indicated by versus; vs.) the general cell-type score. The relative score for exhausted CD8 + T-cells over the total CD8 + T-cells was significantly higher in the responders compared to non-responders ($p = 0.02$, Mann–Whitney $U$-test), and positively correlated with the percentage of best tumor regression from baseline in patients with clinical benefit ($p = 0.01$, Spearman's correlation, Rho 0.68; $n = 13$). The cell-type scores for total CD8 + T-cells or the relative cell-type score for Exhausted CD8 + T-cells over the total tumor infiltrating lymphocytes (TILs) were not significantly different in the responders compared to non-responders. **e** Immune score associated with objective response ($p = 0.01$, Fisher's exact test). **f** Waterfall plot of best percent of tumor regression from baseline as annotated by Sig3 and Immune score (IS) showing that all patients with tumor regression (dashed line represents ≥30%) were positive for Sig3, Immune score or both. **g** Combined score of tumors being positive for Immune score, Sig3 or both associated with clinical benefit ($p = 0.01$, Fisher's exact test). **h** Positivity the combined score significantly correlated with prolonged PFS ($p = 0.002$, Log-rank test). **i** None of the patients whose tumors were negative for the combined score achieved objective response ($p = 0.06$, Fisher's exact test). All test were two-sided. No adjustment was made for multiple hypothesis testing (see materials and methods). Box plots are presented as the range (whiskers), center line as the median, bounds of box mark the highest and lowest quartiles, and the dashed line represents the mean.

clustering of the cell types using 10 nearest neighbors, the neighborhoods with the highest scores for exhausted CD8 + T-cells involved also PD-L1 + macrophages and dendritic cells (Fig. 4e, f). This finding suggests that the interaction between macrophages or dendritic cells and exhausted CD8 + T-cells may be the most relevant cell-cell interaction for the PD-1/PD-L1 mediated immune suppression in this patient.

The second extreme responder in the trial had a PR, with a 53% tumor regression from the baseline and a durable response of over one year and ongoing at the time of the data cutoff. Her tumor was also positive for Sig3, and OncoPanel sequencing revealed high-level genomic amplifications in *CD274* (PD-L1) and *PDCD1LG2* (PD-L2), which were confirmed by FISH (Fig. 4g). The tCyCIF quantitative single-cell analysis revealed that neutrophils, antigen presenting cells and macrophages had the highest PD-L1 expression (Supplementary Fig. 4E). Neighborhood analysis showed increased proximity of the CD8 + T-cells and the PD-L1-positive tumor cells, whereas the PD-L1-positive macrophages clustered separately (Fig. 4h). Further the exhausted CD8 + T-cells spatially clustered together with the PD-L1 + tumor cells whereas the neighborhoods with the PD-L1-positive macrophages clustered spatially separately, with a low neighborhood score for the exhausted CD8 + T-cell (Fig. 4i). Unlike the first extreme responder in which exhausted CD8 + T-cell preferentially were adjacent to PD-L1 + macrophages and dendritic cells, data from this patient therefore suggested enriched immune suppressive PD-L1/PD-1 signaling specifically between the PD-L1-positive cancer cells (which exhibited *PD-L1/PD-L2* amplification) and exhausted CD8 + T-cells.

To confirm these findings we performed 12-marker tCyCIF on regions of interest at higher magnification; this type of imaging is increases the resolution >3-fold (Fig. 4j and Supplementary Table 2). In the first patient (first row), we confirmed high-level PD-L1 expression in the macrophages, which were next to exhausted CD8 + T-cells, with strong co-staining of PD-1 and PD-L1. By contrast, in the second patient with *PD-L1* amplified tumor (second row), there was a clear PD-L1 staining in the tumor cell compartment, and the exhausted CD8 + T-cells were in closer proximity to the tumor cells than to macrophages.

**Exhausted CD8 + T-cell interactions correlate with response.** We applied a statistical neighborhood analysis the t-CyCIF data of all the 26 samples to determine whether spatial interactions of cell subpopulations in the tumor microenvironment associate with response. We compared the frequencies of spatial interactions between the different cell types and functionally relevant cell states to a control group of 1,000 random permutations of all cell labels, with a significant ($p < 0.05$) positive fold-change indicative of attraction and a negative fold-change indicative of avoidance (Supplementary File). Since exhausted CD8 + T-cells were identified as the key cell-type associated with response, we next focused on the spatial interactions of other cell types towards the exhausted CD8 + T-cells. We observed a cluster of significant attractions of exhausted CD8T-cells with macrophages, antigen presenting cells, and T-cell subpopulations (Fig. 5a). Importantly, and a higher attraction between exhausted CD8 + T-cells and macrophages significantly associated with response ($p = 0.02$, Mann–Whitney $U$-test, Fig. 5b), whereas there were no differences in the interactions of exhausted CD8 + T-cells with other cell types in the responders compared to non-responders (data presented in Supplementary File). To look closer into the PD-1/PD-L1 spatial interactions in the tumor microenvironment, we next looked into which of the PD-L1-positive cell subpopulations forms the largest fraction of neighbors of an exhausted CD8 + T-cell in the responders. We found that the responders had a higher fraction of PD-L1-positive macrophages (Fig. 5c) and tumor cells (Fig. 5d) as neighbors of an exhausted CD8 + T-cell compared to the non-responders. Moreover, the fraction of PD-L1-positive tumor cells, and not macrophages (Supplementary Fig. 4F), interacting with exhausted CD8 + T-cells was significantly higher in Sig3-positive tumors compared to Sig3 negative ($p = 0.004$; Mann–Whitney $U$-test, Fig. 5e), indicating that the exhausted CD8 + T-cells are more frequently surrounded by PD-L1-positive tumor cells in Sig3-positive tumors compared to Sig3-negative tumors. A graphical summary of the main findings in the study are presented in Fig. 5f.

## Discussion

We report two previously unrecognized candidate predictive biomarkers to the combination of PARPi niraparib and PD-1 inhibitor pembrolizumab therapy in platinum-resistant ovarian cancer: the presence of mutational signature 3 as a surrogate of HRD and a positive immune score (IS) as a surrogate of interferon-primed, CD8-exhausted effector T-cells in the tumor microenvironment. Presence of one or both tumor features was associated with significantly prolonged PFS (HR = 0.32) while absence of both was associated with absence of response to niraparib/pembrolizumab (ORR 0%). These results are clinically relevant and suggest that Sig3 and immune score may aid in selection of patients with platinum-resistant ovarian cancer who

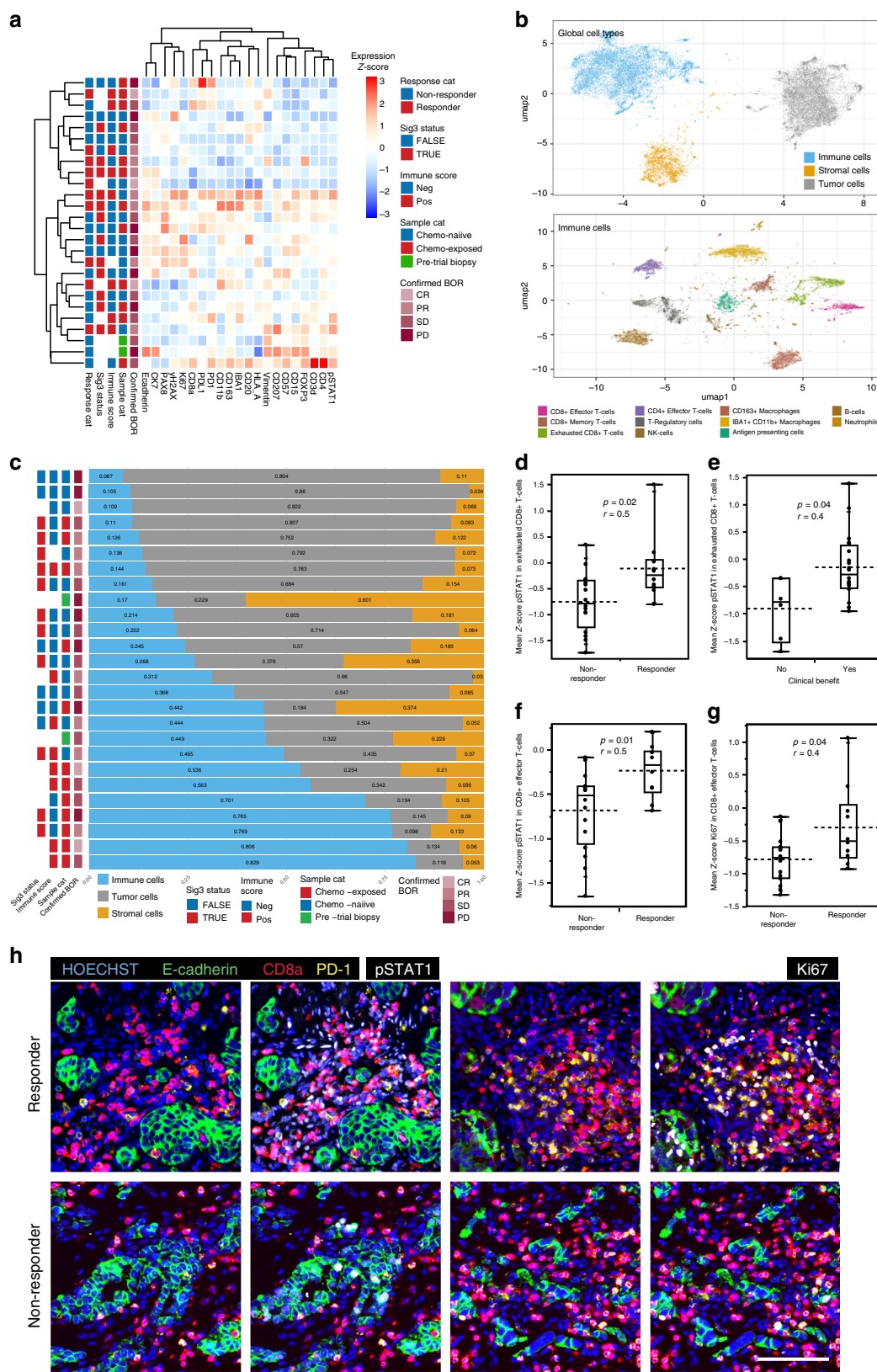

would benefit from niraparib/pembrolizumab. These findings also underscore the translational relevance, given that several previously reported biomarkers of response to PARPi and immune checkpoint inhibitors were unable to predict responses to the niraparib/pembrolizumab combination.

In addition to the assessment of mutations in specific homologous recombination repair genes, such as *BRCA1* and *BRCA2*, several genomic approaches have been utilized to detect HRD in clinical samples. These approaches include detection of hypermethylation of key DNA repair genes such as *BRCA1* and

**Fig. 3 Interferon activation and proliferative state of CD8 + T-cells associate with response. a** A hierarchical clustering heatmap of the mean expression levels of single-cell quantification using tCyCIF and annotated with Sig3 status, Nanostring Immune score, sample category (Sample cat), and confirmed best objective response (confirmed BOR) in 26 patients. **b** Cell-type calls visualized using semi-supervised UMAP dimensionality reduction reveals the clustering of tumor-stroma immune (upper panel), and the tumor microenvironment immune cell subpopulations (lower panel) into distinct clusters. **c** Chemo-exposed samples had a higher immune- and stromal infiltration in the tumor microenvironment. The proportions of immune (blue), tumor (gray) and stromal (yellow) cells of the 26 samples (rows) analyzed via single-cell imaging. **d** Mean pSTAT1 protein expression in exhausted CD8 + T-cells was higher in responders ($n = 10$) compared to non-responders ($n = 16$) and **e** in patients with clinical benefit ($n = 20$) compared to patients with no clinical benefit ($n = 6$). **f** pSTAT1 expression and **g** Ki67 levels in effector CD8 + T-cells associated with response. $p < 0.05$ was considered significant, $r$; effect size, Mann–Whitney $U$-test, all test were two-sided. **h** We observed increased pSTAT1 and Ki67 expression in the exhausted and effector CD8 + T-cells in the responders (upper row) compared to non-responders (lower row). Scale bar 50 μm. Box plots are presented as the range (whiskers), center line as the median, bounds of box mark the highest and lowest quartiles, and the dashed line represents the mean.

*RAD51C*[14] and detection of specific genomic aberrations or genomic "scars" that are characteristic of HRD such as loss of heterozygosity (LOH)[15], telomeric allelic imbalance (TAI)[16] and large-scale state transitions (LST)[16], all of which are captured by the Myriad HRD test (which was evaluated in the TOPACIO trial[7]). Functional biomarkers of HRD have also been developed, such as the assessment of RAD51 foci assembly in cancer cells, which is known to correlate with proficient homologous recombination[4,17–19]. Unlike other markers for HRD, Sig3 is a specific mutational signature characterized by a high number of larger deletions (up to 50 bp) with overlapping microhomology at breakpoint junctions, and commonly detected in ovarian, breast and pancreatic cancers[12]. Its presence reflects the fact that HRD leads to dependence on alternative error-prone repair mechanisms such as alternative non-homologous end joining (alt-NHEJ) or microhomology-mediated end joining, which utilize microhomology at rearrangement junctions to rejoin and repair DNA double strand breaks. Using a machine-learning-based algorithm (SigMA)[11] applied on panel-based sequencing data from our institutional targeted sequencing assay (OncoPanel), we found that Sig3 positivity was significantly associated with clinical benefit and prolonged PFS to the combination of niraparib and pembrolizumab. To our knowledge, this is the first time that Sig3 positivity has been associated with a response to PARPi therapy (either alone or in combination) in ovarian cancer or any other malignancy. Of note, our approach using SigMA facilitated the detection of Sig3 on panel-based sequencing, which is routinely performed on patient samples in clinical practice nowadays, without the requirement for genome-wide sequencing or fresh sample material. Furthermore, Sig3 positivity was identified in a larger proportion of tumors (51% of samples) than positivity for other markers of HRD, suggesting that Sig3 may identify HRD tumors that are missed by other HRD assays. Importantly, no assay is currently considered "gold standard" for assessing HRD in the clinical setting. Consistently, data from two large randomized phase III trials of PARPi maintenance in ovarian cancer, which incorporated either LOH, or the combination of LOH, TAI, and LST as biomarkers of HRD, showed that these measures did not capture all responders to PARPi[20,21]. In fact, PARPi responses occurred even if tumors were negative for any of the HRD biomarkers thereby prompting the FDA to approve PARPis regardless of the status of the explored biomarkers of HRD.

In the context of the TOPACIO study, Sig3-positive status in patients with acquired platinum-resistant ovarian cancer may indicate either non-restored or incompletely restored homologous recombination and ongoing reliance on error-prone DNA repair mechanisms, thus leading to retained sensitivity to PARP inhibition. Additionally, tCyCIF single-cell imaging showed that Sig3-positive tumors exhibited higher PD-L1 expression in tumor cells and in macrophages, suggesting that Sig3 positivity may also be a surrogate of enhanced immunogenicity and thus response to immune checkpoint blockade. This is consistent with the

literature linking HRD with enhanced immunogenicity and increased PD-L1 expression in ovarian cancer[22,23]. Despite being an independent predictor for PFS and a surrogate for both HRD and enhanced immunogenicity, Sig3 positivity did not capture all responders to niraparib/pembrolizumab. Rather, we show that the combination of Sig3 positivity with a positive immune score determined the responses by increased interferon activation and exhausted CD8 + T-cells at the mRNA level. Consistently, using single-cell imaging we revealed an interferon-activated state of exhausted and effector CD8 + T-cells to be significantly associated with response, further highlighting the role of exhausted CD8 + T-cells and the response to this niraparib/pembrolizumab combination.

In the analysis of the two extreme responders, the response in the first patient was attributed to the high proportions and spatial clustering of the PD-L1-positive macrophages and exhausted CD8 + T-cells in the tumor microenvironment. By contrast, the response in the second patient was driven by genomic *PD-L1* and *PD-L2* amplification in the tumor cells, and their spatial interaction towards CD8 + T-cells. To our knowledge, this is the first time that *PD-L1/PD-L2* amplification has been identified in an extreme responder to immunotherapy in ovarian cancer. Besides the two extreme responders, advanced single-cell spatial analyses in the whole dataset revealed prominent spatial interactions of exhausted CD8 + T-cells with macrophages and tumor cells in the responders, suggesting the that both interactions potentially contribute to the CD8 + T-cell exhaustion in the ovarian cancer tumor microenvironment. The finding that tumor cell PD-L1/exhausted CD8 + T-cell interaction occurred particularly in Sig3-positive tumors provides evidence on the spatially divergent tumor microenvironment of HR-deficient tumors, potentially contributing to the enhanced responses seen in Sig3-positive tumors. In this study, the t-CyCIF single-cell imaging provided valuable insights into mechanisms of response to niraparib/pembrolizumab. Overall, our data support the premise that both molecular and spatial features in the tumor microenvironment, including spatial interactions of immune cell subpopulations at different functional states are potentially linked to response to immunotherapy, further underscoring the power of multiplexed imaging in biomarker discovery over conventional methods[24].

We acknowledge certain limitations of this study. The two candidate predictive biomarkers we identified in this Phase 1/2 trial of niraparib/pembrolizumab in ovarian cancer are exploratory and require independent validation in subsequent studies of this or analogous PARPi/PD-1 inhibitor combinations. Samples were collected from patients enrolled in 34 different study sites in the United States, and, despite our best efforts, insufficient tumor material was available for all patients for completion of all aforementioned studies. These limitations notwithstanding, our findings are clinically and translationally-relevant given that combined PARP and immune checkpoint inhibition is an area of active clinical investigation with several ongoing clinical trials in

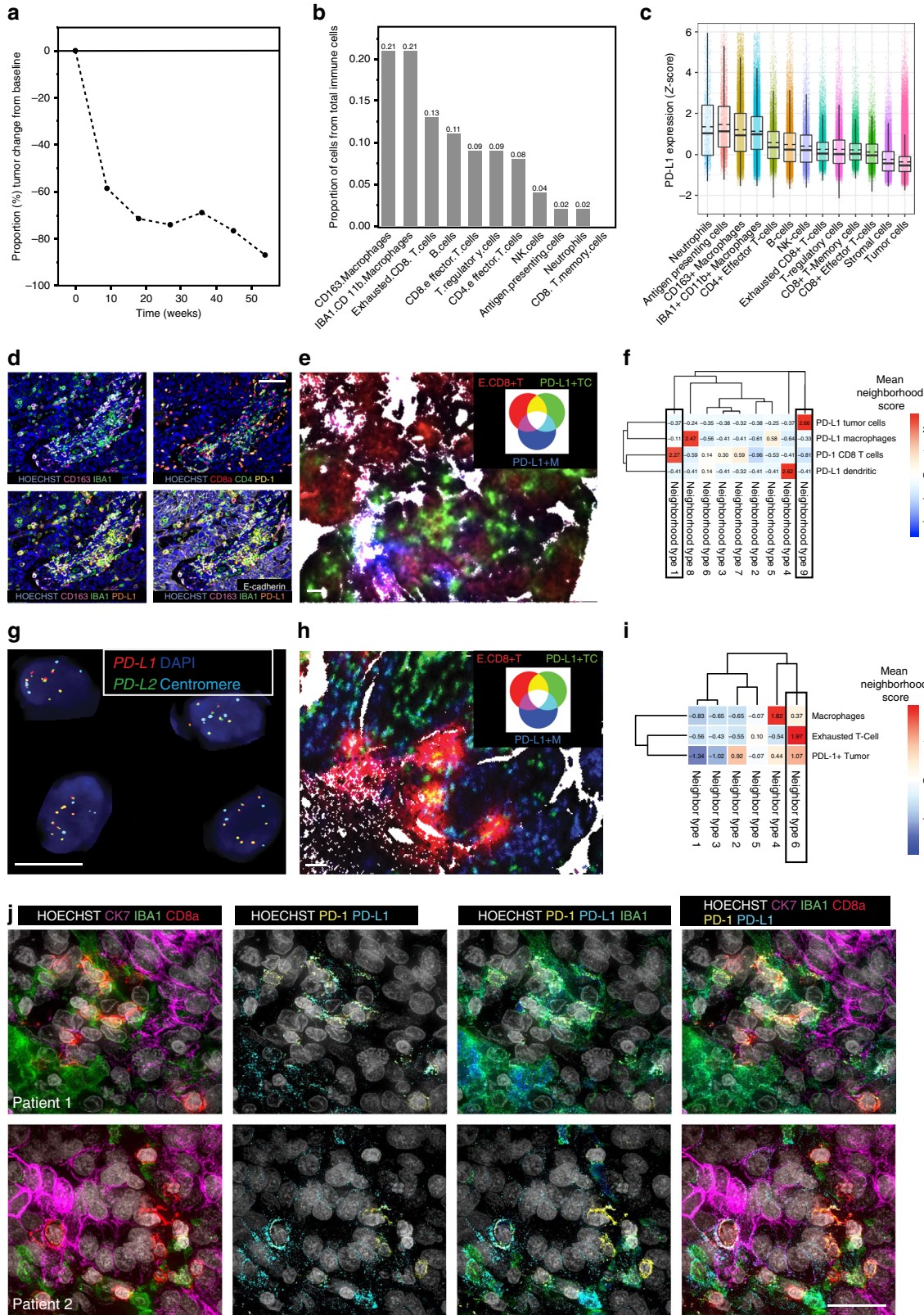

ovarian cancer (in both first-line and recurrent settings), as well as in multiple other cancer types. Our study highlights that careful analysis of genomic information and single-cell spatially resolved data from clinical samples can provide valuable information on the determinants of response to therapy, and accelerate the development of predictive biomarkers to aid in patient stratification.

## Materials and methods

**Tumor samples**. Formalin-fixed paraffin-embedded (FFPE) tumor samples were collected from the 62 patients that had been enrolled from 34 sites into the TOPACIO study[7]. The study was conducted in accordance with ethical principles founded in the Declaration of Helsinki. This study received central approval by the Dana-Farber institutional review board, and/or relevant competent authorities at each site. All patients provided written informed consent to participate in the study. The archival FFPE samples were collected when sufficient (>20% tumor

**Fig. 4 Two extreme responders show differential spatial patterns of cellular interactions in the tumor microenvironment. a** The graph showing the percentage of tumor regression from baseline over time (weeks) in the first extreme responder; she achieved a PR lasting over 10 months with 87% tumor regression from the baseline. **b** The proportion of immune cell subpopulations out of total immune cells in the tumor microenvironment. The patient's tumors immune infiltration was enriched in macrophages and exhausted CD8 + T-cells. **c** The normalized (z-score) PD-L1 expression according to the tumor microenvironment cell subpopulations. The highest PD-L1 expression was observed in CD163 + IBA1 + CD11b + Macrophages. Individual dots represent single cells. Box plots are presented as the range (whiskers), center line as the median, bounds of box mark the highest and lowest quartiles, and the dashed line represents the mean. **d** Multiplexed immunofluorescent images confirmed the high infiltration of macrophages and PD-1 positive exhausted CD8 + T-cells (higher row), and the high expression of PD-L1 in the macrophages (lower row). Scale bar 50 µm. **e** Spatial visualization of neighborhood (k = 10) composition shows increased interaction between PD-1+ exhausted CD8 + T (E.CD8 + T)-cells and PD-L1-positive macrophages (PD-L1 + M) shown in magenta, compared to PD-L1-positive tumor cells (PD-L1 + T) Scale bar 50 µm. **f** K-means clustering indicated that neighborhood clusters containing PD-1+ exhausted CD8 + T-cells contain PD-L1 + macrophages and not PD-L1-positive tumor cells. **g** The second extreme responder exhibits *PD-L1* and *PD-L2* amplification confirmed by FISH. Scale bar 10 µm. **h** Spatial visualization of neighborhood (k = 10) composition shows increased interaction between PD-1+ exhausted CD8 + T (E.CD8 + T)-cells and PD-L1-positive tumor cells (PD-L1 + T) shown in yellow, compared to PD-L1-positive macrophages (PD-L1 + M). Scale bar 50 µm. **i** K-means clustering indicated that neighborhoods containing most of the PD-1+ exhausted CD8 + T-cells cluster together with the PD-L1 + tumor cells and less with the PD-L1-positive macrophages. **j** High-resolution imaging of the two extreme responders using cyclic immunofluorescence. First row depicts the first patient, with exhausted CD8 + T-cells, tumor cells and macrophages in the tumor microenvironment (first column), with positive PD-L1 expression (second column) in the IBA1 + macrophages, most of which were also positive for CD163 (third column), and co-localization of the exhausted CD8 + T-cells with the PD-L1-positive macrophages (fourth column). In the second patient (second row), the exhausted CD8 + T-cells were spatially more next to the tumor cells (first column), while there was clear staining of PD-L1 also in the tumor cell compartment (second column, arrows), in addition to the macrophages (third column), and PD-1/PD-L1 the exhausted CD8 + T-cells. Scale bar 20 µm.

content) tumor material was available. Summary of the clinical data and correlative analyses is presented in Table 1. These samples were obtained either from the time of diagnosis (n = 30, chemo-naive), or after a median of 4.3 months (range 1.5–91.1) following diagnosis, during which time the patient received platinum-based chemotherapeutics as part of clinical care (n = 32, chemo-exposed). No archival resection tissue was available from two patients so biopsies were obtained. The median time from diagnosis to trial entry was 35.8 months (range 11.4–136.5 months). The HRD and clinical data are presented in Supplementary table 4. The timing of the sample did not affect the results from targeted sequencing analyses or RAD51 immunohistochemistry.

**Targeted sequencing**. BRCA mutation testing (tBRCAmut), and HRD testing were previously performed using the Myriad Genetics (Salt Lake City, UT, USA) research assay[7]. BROCA testing was performed as previously described[14], and included sequencing of 84 DNA repair genes and methylation analysis for *BRCA1* and *RAD51C*. In addition, OncoPanel sequencing was performed at the Dana-Farber Cancer Institute core[25] on samples from 35 patients, which had >20% tumor purity as assessed by H&E staining. Briefly, the pooled sample reads were deconvoluted and sorted using the Picard tools, and the reads were aligned to the reference sequence b37 edition from the Human Genome Reference Consortium, using bwa (version0.7.17). Duplicate reads were identified and removed using Picard (version 1.90). The alignments were further refined using the Genome Analysis Toolkit (GATK, version 1.6-5-g557da77) for localized realignment around indel sites and recalibration of the quality scores. Mutation analysis for single-nucleotide variants was performed using MuTect v. 1.1.4 and annotated by Oncotator, and insertions and deletions were called using Indelocator. For each sequencing run, non-neoplastic FFPE and blood samples were included to identify and filter batch-specific sequencing artefacts. To filter out potential germline variants, the standard pipeline removes SNPs present at >0.1% in Exome Variant Server, NHLBI GO Exome Sequencing Project (ESP), Seattle, WA (URL: http://evs.gs.washington.edu/EVS/ accessed 30 May, 2013), present in dbSNP, or present in an in-house panel of normals (n = 141), but rescues those also present in the COSMIC database. We further filtered this data by removing variants present at >0.1% in the gnomAD v.2.1.1 database or were annotated as Benign or Likely Benign in the ClinVar database. Any filtered variants that were reported in COSMIC more than twice were rescued and presented for manual review. All remaining mutations were used for the mutational signature calls with SigMA, including mutations that do not have a consequence such as synonymous and intronic mutations. Mutational signatures were called using SigMA[11]. For this study, SigMA was optimized for OncoPanel using simulations generated specifically for its library using whole genome sequenced ovarian cancer datasets[11]. To minimize false positives, we applied a stringent threshold so that the estimated false-positive rate was 2%, which corresponds to a sensitivity of 65%. The validation of Sig3 calls is show in Supplementary Fig. 1. The SNV and SCNA calls from the International Cancer Genome Consortium (ICGC) project were downloaded from its DCC data portal (https://dcc.icgc.org/releases). Consensus SNV and SCNA calls for the MSK-IMPACT panel data[26] were downloaded from the cBioPortal (http://cbioportal.org/msk-impact). SCNA calls for the MSK-IMPACT data were produced using CNVkit[27]. For two patients analyzed for Sig3 the data for best objective response was not available.

**RAD51 IHC assay, and fluorescent in-situ hybridization (FISH) for PD-L1 and PD-L2**. Immunohistochemistry (IHC) for RAD51[8], and FISH for *PD-L1* and *PD-L2* was performed on tumor sections[28].

**NanoString mRNA expression analysis**. Total RNA was isolated from 2 to 4 5-µm FFPE sections with a Qiagen total RNA kit (Cat# 75144), and quantified by Bioanalyser. The NanoString assay was performed using the PanCancer IO 360 Gene Expression Panel with an additional 30 DNA repair genes as spike-ins (Supplemental Table 1). Gene expression was normalized to 20 housekeeping genes. The data was analyzed using the NanoString NSolver Advanced Analysis platform. Pathway- and cell-type scores were calculated as the first principal component of pathway gene normalized[29,30]. Immune score positivity was calculated from the pathway and cell-type scores using a cutoff of the ≥25% as positive for the score, and below that as negative. In the chemo-naive samples IS was called positive if the sample presented with a score of ≥25% of any of the three interferon pathways due to the overlap of the genes in the three pathways. In the chemo-exposed samples, a positive IS was assigned to a sample if the exhausted CD8 + T-cell/CD8T-cell score was among the ≥25%. A combined score was assigned using Sig3 and IS data so that tumors positive for IS, Sig3, or both being positive, and tumors, which were negative for both IS and Sig3 being negative for the combined score.

**Tissue-based cyclic multiplexed immunofluorescence (tCyCIF)**. The samples were stained with the validated antibodies (Supplementary Table 2) and scanned with RareCyte CyteFinder scanner following the tCyCIF protocol[13]. Scanned image files were corrected using the BaSiC tool, and stitched and registered using the ASHLAR algorithm[31] to align image tiles and successive images of tiles from all cycles to each other. Cell segmentation was performed by applying marker-controlled watershed segmentation to pixel probability maps generated with a UNet neural network[32]. Median fluorescence intensities were computed for each cell and each channel with HistoCAT v1.73[33]. Poor quality events were filtered out based on loss of signal across cycles, background signal from the initial cycle, and solidity metrics.

**Cyclic immunofluorescence in high-resolution imaging**. Z-stacks of 5 µm tissue were acquired on a Deltavision Elite (GE Life Sciences) using a 60×/1.42NA objective lens with oil matching for spherical aberration correction. Excitation channels were 632/22 nm (peak emission/half-width; nominally Cy5), 542/27 nm (TRITC), 475/28 nm (FITC), and 390/18 nm (DAPI) in that sequence on an Edge 5.5 sCMOS camera. Z-stacks were deconvolved using the constrained iterative algorithm in SoftWorx, maximum intensity projected and cycles then registered with DAPI channel using MATLAB (version 2018b, The MathWorks, Inc., Natick, Massachusetts, United States.).

**Cell class-based analysis of single-cell imaging data**. We computed cell-type labels for each cell using the R package flowSOM[34] Briefly, we clustered all cells into a 100-node self-organized map using markers that were used to annotate the cell types and then computed a score for each node and each cell-type label with the flowSOM function *QueryStarPlot*. This process was performed first for global cell types defined as stromal cells, tumor cells, and immune cells, followed by

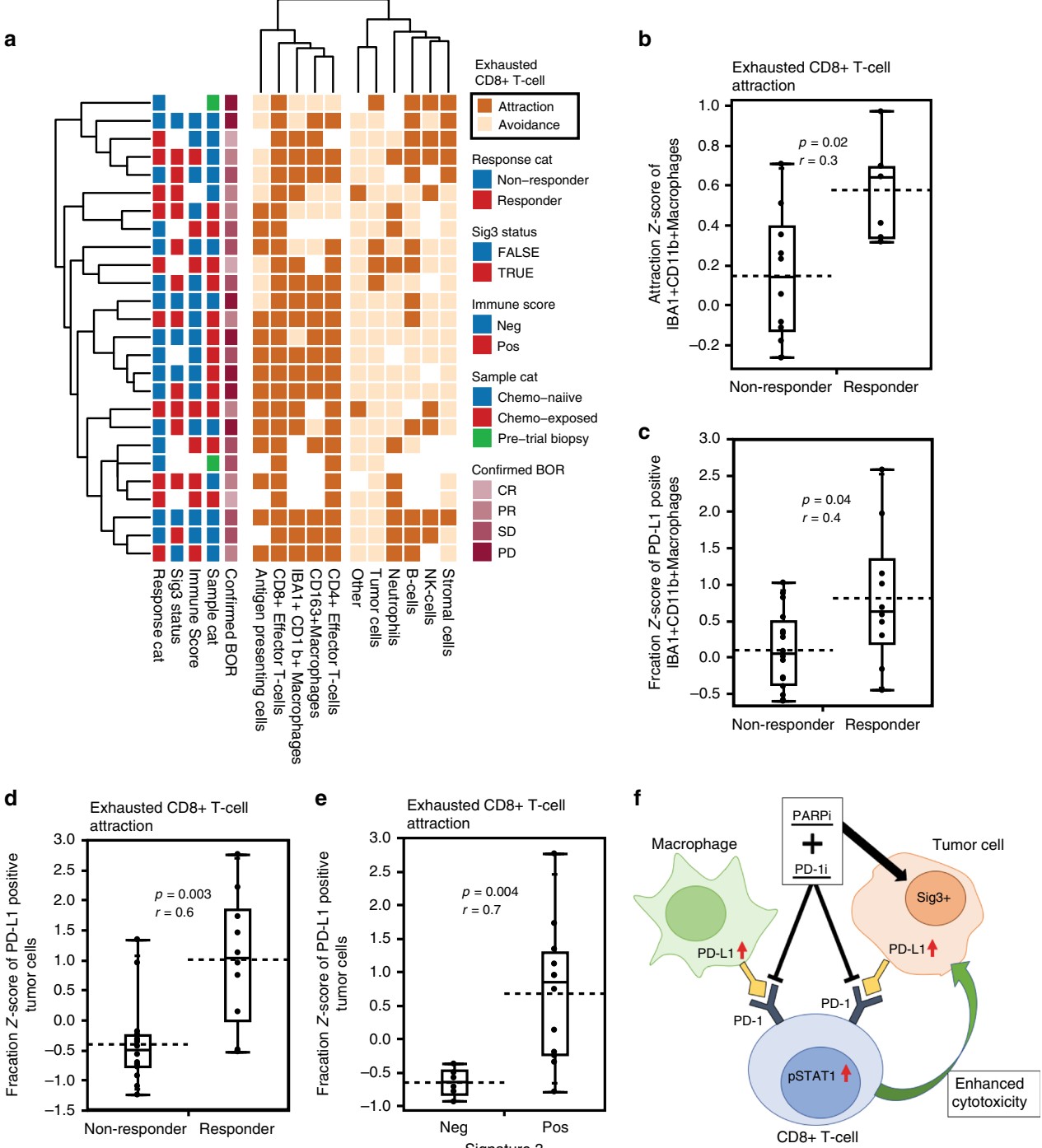

**Fig. 5 Spatial interactions of exhausted CD8 + T-cells associate with response and mutational signature 3. a** A hierarchical clustering of significant interactions of other cell types towards exhausted CD8 + T-cells from neighborhood analysis using permutation test; positive fold-change being attraction and a negative as avoidance, with a two-sided test and a cutoff of $p < 0.05$. **b** In the 16 tumors with significant attraction, of IBA + CD11b + Macrophages towards exhausted CD8 + T-cells the attraction score was higher in the responders ($n = 6$) compared to non-responders ($n = 10$). The responders ($n = 10$) had a significantly higher fraction (z-score) of PD-L1-positive macrophages (**c**) and tumor cells (**d**) interacting with exhausted CD8 + T-cells compared to non-responders ($n = 16$). **e** The fraction (z-score) of PD-L1-positive tumor cells neighboring exhausted CD8 + T-cells was higher in the Sig3 + ($n = 12$) compared to Sig3– ($n = 6$) tumors. $p < 0.05$ was considered significant, $r$; effect size, Mann–Whitney U-test. **f** The model summarizing the findings in the present study. The responses to the combination of niraparib and pembrolizumab are determined by tumor mutational signature 3-positivity, associated with increased PD-L1 positivity in the macrophages and tumor cells, the interferon-primed exhausted CD8 + T-cells and activated effector CD8 + T-cells, and their spatial interactions in the tumor microenvironment. Box plots are presented as the range (whiskers), center line as the median, bounds of box mark the highest and lowest quartiles, and the dashed line represents the mean.

recursive iterations within each cell subtype represented by the antibody panel (Supplementary Table 7). To visualize the data, we applied semi-supervised UMAP dimensionality reduction using cell-type calls as the target metric (https://arxiv.org/abs/1802.03426). The cell labels were then used to identify relevant spatial interactions between cell types and reveal cellular organization using a permutation test[33]. We compared the frequencies of interactions between all pairs of cell types to a control group of 1000 random permutations of all the cell-type labels. An interaction was identified as "attraction" or "avoidance" depending on the sign of the fold-change between actual frequencies to those from the randomized scenarios. Significance threshold was set at $p < 0.05$. To compare the abundance of PD-L1 + neighbors of Exhausted T cells in each sample, we gated PD-L1 + cells within each sample manually and calculated the fraction of PD-L1 + neighbors to Exhausted T cells. Sample-wise z-score normalization is used to visualize and compare between samples.

**Statistics**. The differences between categorical variables were tested using the Fisher's exact test, and continuous variables were compared between groups using Mann–Whitney $U$ or Student's $t$-test when appropriate. Continuous variables were correlated with linear regression and Spearman's Rho. Kaplan–Meier graphs were plotted using standard methodologies and patients that remained on treatment without progression were censored. Outcomes were compared using the log-rank test. No patients were lost to follow-up. Cox-proportional hazards models were conducted using the likelihood ratio Chi-square test. The key analyses are based on <20 simultaneous statistical tests (<20), and thus correcting for multiple hypothesis testing was not considered required. For Nanostring pathway analyses, multiple hypothesis correction was not performed due to violation of due to the violation of the assumption of independence due to overlapping genes in the signatures. Effect size ($r$) for Mann–Whitney $U$-statistics was calculated as the $Z$-statistic divided by the square root of the number of samples. As the sample material from the clinical trial were extremely limited, we performed each experiment only once (sequencing, Nanostring, tCycIF staining), or twice (the high-resolution imaging). Two-sided $p$-value <0.05 was considered significant.

**Reporting summary**. Further information on research design is available in the Nature Research Reporting Summary linked to this article.

## Data availability

The source data underlying Figs. 1, 2, 3, and 5 are provided as a Source Data file. Oncopanel sequencing, Nanostring and the highly multiplexed images are available in Synapse at https://doi.org/10.7303/syn21593960.

## Code availability

Code and algorithms produced in this study for the multiplexed imaging data analysis are available at https://github.com/farkkilab/pubs/tree/master/Farkkila-et-al-1.

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

## Acknowledgements

We thank the patients and their families for their participation in this study, as well as the study teams at each of the study sites. Research supported by the Stand Up To Cancer-Ovarian-Cancer Research Alliance—National Ovarian Cancer Coalition Dream Team Translational Research Grant and a SU2C Catalyst® Research Grant also supported by a charitable donation from Merck, (SU2C-AACR-DT16-15). Both grants are administered by the American Association of Cancer Research, the Scientific Partner of SU2C. Stand Up To Cancer is a division of the Entertainment Industry Foundation, a 501(c)(3). This study was also supported by Ovarian Cancer Research Fund, and the TESARO, GSK

Company, and Merck & Co. The aforementioned funding agencies/companies have not participated in the study design or analysis of the results. This research was also supported by the Ludwig Center at Harvard and U54-CA225088 to Peter Sorger, BioEntrepreneur-Fellowship of the University of Zurich, reference no. BIOEF-17-001 and Early Postdoc Mobility fellowship (no. P2ZHP3_181475) to Denis Schapiro.

## Author contributions

A.F. coordinated, performed, and analyzed all the correlative data and wrote the manuscript; D.G.C. and P.J.P. performed the mutational signature analysis; J.C. generated and analyzed the tCycIF multiplexed imaging data; C.A.J. performed the tCycIF and high-resolution multiplexed imaging; H.N. analyzed the Nanostring data; B.K. performed and analyzed the RAD51 IHC; Z.M. optimized the antibodies for tCycIF; C.Y. performed cell segmentation and high-resolution multiplexed imaging; Y.A.C. and D.S. generated the single-cell quantitated imaging data; Y.Z. coordinated the processing and distribution of the clinical trial samples; J.R.G. provided the clinical data for the trial; B.J.D. and P.M. coordinated and performed the clinical trial; S.S. evaluated the histology of tumor sections from the clinical trial; E.G. performed the OncoPanel sequencing on the samples; S.R. and A.L. validated and performed the FISH; D.C., G.I.S., and U.A.M. supported and coordinated the clinical trial; S.H. supervised J.C. and the data analysis; P.K.S. developed the tCycIF technology and data analysis; E.M.S. performed and interpreted the BROCA sequencing, A.D.A. and P.K. conceived and supervised the correlative analyses and wrote the manuscript. All authors contributed to the writing and editing of the manuscript.

## Competing interests

J.G. and Y.Z. were previously employed by TESARO, and now is a GSK employee. B.D. was an employee of Tesaro at the time of the work related to this manuscript. G.I.S. receives research funding from Merck & Co. S.S. is a consultant for RareCyte, Inc. P.K.S. is a member of the SAB or Board of Directors of Applied Biomath and RareCyte Inc and has equity in these companies. In the last 5 years, the Sorger lab has received research funding from Novartis and Merck. Sorger declares that none of these relationships are directly or indirectly related to the content of this manuscript. For the SigMA algorithm, a patent application titled "Systems and methods for classifying tumors" has been filed under the Patent Cooperation Treaty (PCT) by authors D.G. and P.P. on 18 September, 2019 following a provisional patent application filed on 24 September, 2018. U.M. has served as a consultant for Merck and has been the North America PI of the NOVA study funded by TESARO. P.K. and A.D.D'A. have served as consultants/members of advisory boards for Merck and Tesaro/GSK. Other authors have no competing interests to declare.
