## [Peer Review File · Nature Communications]

Reviewers' Comments:

Reviewer #1:

Remarks to the Author:

This is a well-conducted correlative study of an immunotherapy trial in ovarian cancer. It is highly descriptive, but novel. It is significantly limited by both sample-size and lack of generalization, but is generally very well written, does not over-reach in its claims. There are three issues that need addressing prior to publication, in my view.

1. HRD Signature Inference

It is critical to validate the inference of mutational signature activity, essentially all discrepancies between this feature and HRD activity can be explained by FPs and FNs in signature inference from the small sequencing panel (which is stated as 447 alterations rather than 447 genes, in which case accurate inference would be remarkable). Full validation (e.g. using WGS in a subset of cases) is critical since this is driving a major conclusion of this study. Along similar lines, please demonstrate that the inferred tumour point mutational burden is in the range of previous sequencing studies of ovarian cancer.

2. Computational Methods

* Please outline a reproducible procedure used for somatic variant detection, the current methods do not have sufficient detail.

* Please specify in the results at first presentation exactly how many tumours were profiled using each method, and provide a clear supplementary table showing the overlap of datasets

* Please report precise statistics with both effect-sizes and p-values throughout (e.g. six pathways specifically enriched among tumours with objective response -- what was the threshold used, what types of effect-sizes were present)

* Please quantify the "clear clustering of the tumour, immune and stromal cells and further to immune subpopulations".

3. Data Availability

All sequencing (raw and variant call) data and NanoString data need to be deposited into a suitable public repository (e.g. dbGaP) as it serves as the central value of this study and this is of course in line with NPG publication guidelines.

Reviewer #2:

Remarks to the Author:

Farkkila and colleagues present an integrated genomic (mutation) and immunomics (IO sequencing and single cells staining) of tumor tissue from patients that had participated in a clinical trial testing the efficacy of combination PARP inhibitor (niraparib) and immune checkpoint blockade (pembrolizumab). Specimens from patients were subjected to extensive testing with a variety of cutting edge modalities. The ultimate conclusion is that a combination of mutation testing and immune response signals identifies all patients that demonstrated improved outcomes as a result of treatment. Additional information from immunocytofluorescence studies show interactions between PD-L1+ macrophages, PD-L1 tumor cells and CD8 T cells. Lastly, 2 extreme responders were characterized showing differential responses. Overall, while there is no independent validation, the data are novel and support the overall conclusions. There are a number of problems, however, that should be considered prior to publication.

1. The introduction is weak with respect to the rationale for the combination. Additional detail about the prior preclinical work that demonstrates synergistic interactions between PARPi and immune checkpoint blockade should be discussed.

2. There should be a table that summarizes the patient population examined in the present study, which is typically standard in these translational publications.

3. Since there are a number of tests and each test was applied to only some patients, the authors

should revise the methods to make it clear how many patients were evaluated with each lab test. The authors should also clarify why some analyses used differing numbers of patients. For example, in Fig 1B the SigMA (which is assumed to be Sig3) is on 37 patients whereas Fig 1D analyzes 39 individuals.

4. Additional cox regression analyses should be considered for the progression free survival studies to determine if the signatures are independent predictors.

5. In general, the font sizes throughout are heterogeneous and some are very difficult to read.

6. The figure legends should include the numbers of patients analyzed.

7. In Figure 2 and other similar figures, it is unclear what the term 'versus' means. This term needs to be described in context. Is it the ratio?

8. Figures 3H and 5F need to be mentioned in the Results section.

9. Figure 2 seems to be missing the Chi-square table for Sig3.

10. The headers for the supplementary tables are cut off in some places. Additional attention should be paid to these tables if they are to be published.

11. The title should be changed to reflect the combination of both HRD and immunomics profiles.

Reviewer #3:

Remarks to the Author:

The manuscript by Farkkila, et al reports on a previously unrecognized candidate predictive biomarkers to the combination of a PARP and PD-1 inhibitor in platinum resistance ovarian cancer patients. These biomarkers may aid in patient selection for future clinical trialing. During the discovery of these biomarkers, substantial effort was put into both genomic analysis of the tumors as well as spatial analysis of the antibody stained biomarkers, providing this novel insight. Overall, this is a novel finding, however the paper needs some revisions prior to consideration for publication explained as follows.

The introduction is focused on a specific type of immunotherapy without any context as to what percentage of ovarian cancer patients this type of therapy would serve. Some additional context for the motivation behind this study in ovarian cancer patients should be added such as what percentage of clinical trial work for ovarian cancer is immunotherapy? How does it compare to other therapies under investigation, etc.

The tables in Figure 2, specifically Figs 2E, 2G and 2I are very large compared to the font size on many of the other elements in the figure. This figure should be rearranged to use the white space in these tabular pieces to increase the size of the font on the axes of the graphs, which are very small and not legible unless magnified to a large font size.

The markers stained by cyCIF in Fig 3H should be denoted in some other way, as it is, it is quite difficult to read since they are overlaid on top of positive staining that obscures the label. This is also a problem in Figure 4D.

In Figs 4 and 5, the extreme response of two patients is highlighted, where the first panel shows the response, but it isn't clear what is being measured?

As an overall comment, the number of acronyms used in the manuscript is high and could easily be reduced, this would substantially improve the readability for an educated scientific audience that are not experts in the field.

Rebuttal letter NCOMMS-19-539397, entitled “Immunogenomic profiling determines responses to combined PARP and PD-1 inhibition in ovarian cancer.”

Author answers are marked with >

Reviewer #1;

1. HRD Signature Inference

It is critical to validate the inference of mutational signature activity, essentially all discrepancies between this feature and HRD activity can be explained by FPs and FNs in signature inference from the small sequencing panel (which is stated as 447 alterations rather than 447 genes, in which case accurate inference would be remarkable). Full validation (e.g. using WGS in a subset of cases) is critical since this is driving a major conclusion of this study. Along similar lines, please demonstrate that the inferred tumour point mutational burden is in the range of previous sequencing studies of ovarian cancer.

> We agree with the referee that mutational signature 3 is a critical delineator of conclusions in the study, and we have now added new figures and detailed description of the Whole Genome Sequencing (WGS) validation for this inference. The presence of mutational signature 3 was inferred from a sequencing panel of 447 genes, and we have now clarified this in the manuscript (page 5, line 177). We performed validation of Sig3 inference using available WGS data of HGSOV from the ICGC consortium (Supplementary Figure 1). Firstly, the tumor mutational burden estimated based on our panel sequencing data was highly consistent with WGS and publicly available panel sequencing data from Memorial Sloan Kettering (Supplementary Figure 1 C). In the present study, we used a strict threshold for the SigMA score to infer Sig3 positivity; this strict threshold corresponds to false positive rate of 2%, and a sensitivity of 65% as presented in Supplementary Figure 1 D. We further confirmed that the panel sequencing simulations from WGS faithfully correspond to the real-life OncoPanel data via plotting of the density of SNVs against the probability of Sig 3 in both data types (Supplementary Figure 1 E). Further, the SigMA scores as well as the number of SNVs show high concordance with simulations from WGS and OncoPanel data in the Sig+ and Sig- groups (Supplementary Figure 1 F-G). The average mutational pattern and fraction of signature 3 among all mutations are compared for WGS data, panel simulations and real-life Oncopanels (Supplementary Figure 1H). The fractions of mutational signatures calculated with non-negative least squares (NNLS) also demonstrate a highly similar contribution of Sig3 in the Sig3+ group, and consistently the lack of Sig3 contribution in Sig3- groups, between the simulated panels and Oncopanels. These extensive validation steps demonstrate a highly reliable mutational signature calling with SigMA.

Unfortunately, there is no additional tumor or DNA left from the samples to perform further sequencing comparison in this study. However, the extensive description of the SigMA method (1) and the validation we have performed here strongly indicate that the Sig3 inference from OncoPanel using SigMA is reliable and faithfully recapitulates the inference from WGS.

2. Computational Methods

*** Please outline a reproducible procedure used for somatic variant detection, the current methods do not have sufficient detail.**

> We have now added this into the methods section (Pages 18-19, lines 516-532).

*** Please specify in the results at first presentation exactly how many tumours were profiled using each method, and provide a clear supplementary table showing the overlap of datasets**

> We have now added these data into a new Table 1 A-B.

*** Please report precise statistics with both effect-sizes and p-values throughout (e.g. six pathways specifically enriched among tumours with objective response -- what was the threshold used, what types of effect-sizes were present)**

> These are now added effect size calculations into all the relevant graphs, and explained the thresholds in the figure legends.

*** Please quantify the "clear clustering of the tumour, immune and stromal cells and further to immune subpopulations".**

> We have now corrected this sentence to further clarify the methodology (page 9, lines 269-270).

3. Data Availability

All sequencing (raw and variant call) data and NanoString data need to be deposited into a suitable public repository (e.g. dbGaP) as it serves as the central value of this study and this is of course in line with NPG publication guidelines.

> We are now in the process of submitting the Oncopanel and Nanostring data to dbGAP, and it will be made available upon publication.

Reviewer #2 (Remarks to the Author):

1. The introduction is weak with respect to the rationale for the combination. Additional detail about the prior preclinical work that demonstrates synergistic interactions between PARPi and immune checkpoint blockade should be discussed.

> We have now added a paragraph to the introduction describing the rationale of the combination including references to the preclinical work done in this context including appropriate references (page 3, lines 85-94).

2. There should be a table that summarizes the patient population examined in the present study, which is typically standard in these translational publications.

> We have now added this as a table to the manuscript (Table 1 A).

3. Since there are a number of tests and each test was applied to only some patients, the authors should revise the methods to make it clear how many patients were evaluated with each lab test. The authors should also clarify why some analyses used differing numbers of patients. For example, in Fig 1B the SigMA (which is assumed to be Sig3) is on 37 patients whereas Fig 1D analyzes 39 individuals.

> We have now added this as a table to the manuscript (Table 1 B). The discrepancy in Fig 1B in the correlation of Sig 3 and response is due to the fact that for two patients analyzed for Sig3 the clinical data for confirmed best objective response was not available. We have now added this information into the Methods section (page 19, lines 542-543).

4. Additional cox regression analyses should be considered for the progression free survival studies to determine if the signatures are independent predictors.

> We have now performed additional Cox regression analyses and show that Signature 3 was indeed an independent predictor of PFS (Supplementary Table 5). This has also been added to the manuscript results (page 10, lines 307-308) and discussion 16 (441 -442).

5. In general, the font sizes throughout are heterogenous and some are very difficult to read.

> We have improved this by unifying the font sizes in the figures.

6. The figure legends should include the numbers of patients analyzed.

> These data have now been added to the figure legends in the cases where the numbers of patients analyzed was not indicated in the figure.

7. In Figure 2 and other similar figures, it is unclear what the term 'versus' means. This term needs to be described in context. Is it the ratio?

> This is the inferred relative cell-type score from Nanostring advanced analysis. We have now further clarified this in the figure legend (page 26, lines 746-746).

8. Figures 3H and 5F need to be mentioned in the Results section.

> These have now been added to the manuscript results section (page 10, lines 299 – 301, and page 13, lines 379-380)

9. Figure 2 seems to be missing the Ch-square table for Sig3.

> The Chi-square table for Sig3 and clinical benefit is presented in Figure 1. The Chi-square table for Sig3 and objective response has now been added to Figure 2 as per reviewers suggestion.

10. The headers for the supplementary tables are cut off in some places. Additional attention should be paid to these tables if they are to be published.

> We have now edited this to ensure the graphical outputs of the files submitted via the submission portal.

11. The title should be changed to reflect the combination of both HRD and immunomics profiles.

> While we acknowledge that this is a reasonable suggestion, we would prefer keeping the title as it is. However, we do remain flexible if the reviewers and the editor insist on this point.

Reviewer #3 (Remarks to the Author):

The introduction is focused on a specific type of immunotherapy without any context as to what percentage of ovarian cancer patients this type of therapy would serve. Some additional context for the motivation behind this study in ovarian cancer patients should be added such as what percentage of clinical trial work for ovarian cancer is immunotherapy? How does it compare to other therapies under investigation, etc.

> We have now added a paragraph to the introduction (page 3, lines 84-93).

The tables in Figure 2, specifically Figs 2E, 2G and 2I are very large compared to the font size on many of the other elements in the figure. This figure should be rearranged to use the white space in these tabular pieces to increase the size of the font on the axes of the graphs, which are very small and not legible unless magnified to a large font size.

> We have now edited this Figure to improve legibility.

The markers stained by cyCIF in Fig 3H should be denoted in some other way, as it is, it is quite difficult to read since they are overlaid on top of positive staining that obscures the label. This is also a problem in Figure 4D.

> We have now edited the legends in these Figures.

In Figs 4 and 5, the extreme response of two patients is highlighted, where the first panel shows the response, but it isn't clear what is being measured?

> We have now further explained this in the Figure legend (page 27, line 784).

As an overall comment, the number of acronyms used in the manuscript is high and could easily be reduced, this would substantially improve the readability for an educated scientific audience that are not experts in the field.

> We have now reduced the number of acronyms to the minimum to increase the readability of the manuscript.

References

1. Gulhan DC, Lee JJ, Melloni GEM, Cortes-Ciriano I, Park PJ. Detecting the mutational signature of homologous recombination deficiency in clinical samples. Nat Genet. 2019;51(5):912-9.

Reviewers' Comments:

Reviewer #1:

Remarks to the Author:

The authors have fully and clearly addressed my concerns

Reviewer #2:

Remarks to the Author:

The reviewers adequately addressed my concerns. Thank you

Reviewer #3:

Remarks to the Author:

Overall the changes to the manuscript have substantially improved its readability and clarity.

However, I am still confused about my previous comment regarding the extreme response of the two patients highlighted in Figures 4 &5 about what is being measured. The response to my query is supposed be to on page 27, line 784 according to the response to reviewers, but now such line exists on that page. Line 784 does exist in the paper, but is in the captions to the supplementary figures. Please clarify.

Rebuttal letter NCOMMS-19-539397, entitled “Immunogenomic profiling determines responses to combined PARP and PD-1 inhibition in ovarian cancer.”

Author answers are marked with >

In Figs 4 and 5, the extreme response of two patients is highlighted, where the first panel shows the response, but it isn't clear what is being measured?

> **We have now clarified in the Figure, and explained this in the figure legend of Figure 4.**